# The Combined Treatment of Curcumin with Verapamil Ameliorates the Cardiovascular Pathology in a Williams–Beuren Syndrome Mouse Model

**DOI:** 10.3390/ijms24043261

**Published:** 2023-02-07

**Authors:** Noura Abdalla, Paula Ortiz-Romero, Isaac Rodriguez-Rovira, Luis A. Pérez-Jurado, Gustavo Egea, Victoria Campuzano

**Affiliations:** 1Department of Biomedical Sciences, School of Medicine and Health Sciences, University of Barcelona, 08036 Barcelona, Spain; 2Department of Medicine and Life Sciences, University Pompeu Fabra, 08003 Barcelona, Spain; 3Centro de Investigación Biomédica en Red de Enfermedades Raras (CIBERER), ISCIII, 28029 Madrid, Spain; 4Genetics Service, Hospital del Mar & Hospital del Mar Research Institute (IMIM), 08003 Barcelona, Spain

**Keywords:** Williams–Beuren syndrome, curcumin, verapamil, CD mice, cardiovascular disease, oxidative stress

## Abstract

Williams–Beuren syndrome (WBS) is a rare disorder caused by a recurrent microdeletion with hallmarks of cardiovascular manifestations, mainly supra-valvular aortic stenosis (SVAS). Unfortunately, there is currently no efficient treatment. We investigated the effect of chronic oral treatment with curcumin and verapamil on the cardiovascular phenotype of a murine model of WBS harbouring a similar deletion, CD (complete deletion) mice. We analysed systolic blood pressure in vivo and the histopathology of the ascending aorta and the left ventricular myocardium to determine the effects of treatments and their underlying mechanism. Molecular analysis showed significantly upregulated xanthine oxidoreductase (XOR) expression in the aorta and left ventricular myocardium of CD mice. This overexpression is concomitant with increased levels of nitrated proteins as a result of byproduct-mediated oxidative stress damage, indicating that XOR-generated oxidative stress impacts the pathophysiology of cardiovascular manifestations in WBS. Only the combined therapy of curcumin and verapamil resulted in a significant improvement of cardiovascular parameters via activation of the nuclear factor erythroid 2 (NRF2) and reduction of XOR and nitrated protein levels. Our data suggested that the inhibition of XOR and oxidative stress damage could help prevent the severe cardiovascular injuries of this disorder.

## 1. Introduction

Along with a quite characteristic neurocognitive profile and dysmorphic features, Williams–Beuren syndrome (WBS, OMIM 194050) individuals present a generalised arteriopathy associated with hyperplasia of vascular smooth muscle cells (VSMC), disorganised and fragmented elastic fibres, and increased number of lamellar structures [1]. Stenotic lesions are typically seen in the thoracic aorta and pulmonary artery; however, they are also present in other locations, such as the renal artery [2]. High blood pressure and supra-valvular ascending aortic stenosis (SVAS) are the most common cardiovascular manifestations of WBS, affecting more than 75% of patients [2]. Severe SVAS often leads to cardiac hypertrophy increasing the risk of complications such as stroke and sudden death [2,3]. These cardiovascular features of WBS are mainly related to a gene-dosage effect due to hemizygosity of the elastin gene (*ELN*) which encodes the elastin protein; however, the phenotypic variability among WBS patients indicates the presence of important modulators of the clinical impact of elastin deficiency [1]. The complete deletion (CD) mouse model, carrying the most common deletion found in WBS patients (from *Gtf2i* to *Fkbp6*), presents a pathological cardiovascular phenotype very similar to that seen in patients with SVAS, hypertrophic cardiopathy and hypertension [4,5]. Moreover, the CD mouse model is the most suitable for assaying potential new therapeutic interventions [6,7].

Oxidative stress is a significant factor contributing to the pathogenesis of several human diseases. It is considered one of the major cardiovascular risk factors impairing endothelial function and inducing abnormal arterial remodelling and vascular inflammation [8]. Oxidative stress is characterised by the overproduction of reactive oxygen species (ROS), which can damage some components of the cell, including proteins, lipids, and DNA [9]. In WBS patients, ROS levels have been described as strong determinants of hypertension risk and vascular stiffness [10,11]. These findings were also supported by comparative studies carried out in another murine model defined by the partial distal deletion (DD) of the WBS orthologous region [12]. Of note, cardiovascular pathology reported in the CD murine model is much less severe compared with those of the DD model. This fact was explained in part by the decreased tissular expression in CD mice of neutrophil cytosolic factor 1 (*Ncf1*), a gene located next to the deleted interval in these mice [4,7]. However, both the heart and the aorta of the CD model permanently exhibit high levels of oxidative stress [5,7].

Biochemical, molecular and pharmacological studies have shown that xanthine oxidoreductase (XOR) is also a relevant source of ROS in the cardiovascular system [13,14,15]. Although the participation of XOR in WBS has not been reported yet; however, other connective tissue diseases, such as Loeys-Dietz and Marfan syndromes, are associated with increased XOR protein levels in aortic samples [16,17]. Interestingly, ROS generated by hypoxanthine and xanthine oxidase can be inhibited by curcumin (CUR) [18] as well as verapamil hydrochloride (VER) [19].

CUR is a natural yellow pigment that has attracted much attention in recent years owing to its wide spectrum of biological properties, including antioxidant, anti-inflammatory, anti-tumour, and anti-microbial activities [20,21]. CUR improved cardiovascular structure and function, especially with the normalisation of systolic blood pressure (SBP) and collagen deposition in rats with diet-induced metabolic syndrome [22]. In addition, CUR has previously shown cardiac protection against palmitate and high-fat diets, promoting the activation of the nuclear factor erythroid 2 (NRF2) [23].

Voltage-dependent calcium channel blockers, such as VER, have been used for a long time as antihypertensive drugs acting as peripheral vasodilators [24]. VER has also been used in the treatment of other cardiac diseases, including arrhythmias and angina pectoris [25]. In addition, VER has been reported to inhibit ROS production after ischemia–reperfusion of the rat liver [26] and has proven its efficacy in the treatment of oxidative stress-associated damage as a potent NRF2 activator [27].

We previously reported that only the combined (CURVER) and not a single treatment (CUR or VER) significantly improved the cognitive dysfunction of CD mice [6]. We then evaluated whether the combinatorial treatment could also represent an effective pharmacological strategy to treat cardiovascular alterations in CD mice, as well as studying the role of XOR as it is an important source of oxidative stress in the cardiovascular system. We herein report that the chronic combinatorial treatment of both compounds (CURVER) ameliorates the main cardiovascular pathology, and this concomitantly occurs with the normalisation of XOR and oxidative stress-associated reactions.

## 2. Results

### 2.1. Only the Combinatorial Treatment with Curcumin and Verapamil (CURVER) Prevents Hypertension in CD Animals

Systolic blood pressure (SBP) and heart rate in beats per minute (BPM) were determined after four weeks of CUR, VER and combined CURVER treatments in conscious male and female mice using a non-invasive Blood Pressure System as previously described [4]. Males and females were finally analysed together once no significant sex differences were observed (Appendix A).

SBP analysis showed a significant genotype/treatment interaction (F_3, 131_ = 4.032, *p* = 0.0088). In individual comparisons, SBP showed significant differences with respect to genotype in CUR and VER independent experimental groups except in mice co-treated with both drugs (CURVER), who presented statistically significant differences with respect to treatment (*p* < 0.0001) (Figure 1A).

Interestingly, two-way ANOVA analysis revealed significant differences at the genotype level, with an increase in BPM in CD compared with wild-type (WT) animals (F_1, 133_ = 21.49, *p* < 0.0001). This agrees with previously published data that positively correlated heart rate slopes with basal blood pressure in the overall population [28]. We also observed a significant treatment effect (F_3, 133_ = 5.677, *p* = 0.0011) with a reduction in the heart rate of treated mice (Figure 1B).

No treatment had any significant effect on the SBP or BPM of WT mice (Appendix A, WT panels). Detailed statistical analyses are shown in Appendix A.

### 2.2. CURVER Treatment Prevents CD-Associated Aortic Stenosis

Considering the above results, we then only performed CURVER administration experiments. Thus, to evaluate if the improvement in SBP in CURVER-treated CD mice correlated with improved aortic parameters, we performed a detailed histopathological analysis (Figure 2).

In the ascending aortic wall of CD animals, we observed a significant increase in the thickness of the tunica media (effect of genotype: F_1, 30_ = 27.99, *p* < 0.0001) (Figure 2A) together with a significant reduction of the aortic lumen diameter (effect of genotype F_1, 30_ = 42.17, *p* < 0.0001) (Figure 2B). CURVER treatment produced an overall improvement both in the ascending aortic lumen diameter and wall thickness (Figure 2A,B, respectively).

As expected, the amount of elastin (evaluated by autofluorescence emission) in the tunica media of CD animals was significantly lower compared with WT animals (effect of genotype F_1, 30_ = 43.72, *p* < 0.0001) regardless of CURVER treatment (F_1, 30_ = 0.5833, *p* = 0.4510) (Appendix A). We reasoned that the thickening of the tunica media observed in CD animals could be the result of compensatory mechanisms for the elastin deficiency, increasing the number of VSMCs, the amount of collagen, or both. Thus, we next analysed VSMCs density (Figure 2C), observing a significant increase in cell density in the ascending aorta of CD mice (effect of genotype F_1, 20_ = 53.93, *p* < 0.0001). CURVER treatment significantly counteracted this alteration (effect of treatment, F_1, 20_ = 9.485, *p* = 0.0059) (Figure 2C).

Subsequently, we analysed the amount of total collagen in the tunica media by Picrosirius Red Staining. Under polarised light microscopy, green (immature) and red (mature) collagen fibres are visualised, being demonstrative of their different fibre thicknesses and assembly compaction and, therefore, of their degree of maturity [29]. We observed a significant increase in total collagen fibres in the tunica media of CD mice (*p* < 0.0001) (Figure 2D and Appendix A), mainly attributable to green fibres (*p* < 0.0001) (Figure 2E and Appendix A). This remodelling was prevented after CURVER treatment with significant differences between CURVER-treated and untreated CD (CD-vehicle) mice (*p* = 0.0005 for total and *p* = 0.0015 for green) (Figure 2D,E). In the case of red fibres, a significant interaction between genotype/treatment was noted (F_1, 46_ = 8.411, *p* = 0.0057), although without achieving significance in Tukey’s multiple comparisons test (Appendix A).

Detailed statistical analyses are shown in Appendix A.

### 2.3. CURVER Treatment Attenuates CD-Associated Cardiac Hypertrophy

We then studied the effect of CURVER treatment on the most characteristic histopathological cardiac parameters occurring in CD mice. Thus, cardiac hypertrophy was evaluated as the contribution of heart weight to total body weight. As in the SBP evaluation, male and female mice were finally analysed together after verifying that there were no sex differences in the phenotype (Appendix A). The characteristic cardiac hypertrophy observed in CD mice (effect of genotype F_(1, 101)_ = 16.37, *p* = 0.0001) was prevented by chronic CURVER administration, avoiding the increased heart weight (genotype/treatment interaction: F_(1, 101)_ = 20.76, *p* < 0.0001; Figure 3A). The histological analysis (Figure 3B) confirmed that CURVER treatment reduced (*p* = 0.5941) the thickening of the left ventricular (LV)-myocardium wall seen in CD-vehicle mice (*p* = 0.0009) (Figure 3C).

Detailed statistical analyses are shown in Appendix A.

### 2.4. Parallel Reduction Levels of Aortic and Cardiac XOR and Oxidative Stress Markers after CURVER Treatment

Due to the relevant role that oxidative stress plays in the development of cardiovascular disease in WBS, we wanted to investigate whether the mechanism(s) by which CURVER treatment prevented the formation of vascular stenosis and cardiac hypertrophy was related to oxidative stress pathways. XOR, as an important ROS source, plays important roles in a variety of pathophysiological states in the cardiovascular system, including ischemia/reperfusion injury, atherosclerosis, and LV dysfunction after myocardial infarction [14,15]. Endothelial cell-generated nitric oxide easily interacts with XOR-derived anion superoxide (O_2_^−^), forming peroxynitrite (ONOO^−^), which in the endothelium of the tunica intima and VSMCs of the media irreversibly generates reactive nitrogen species residues such as 3′ -nitrotyrosine (3-NT) [30,31], which is usually evaluated as a redox stress marker.

Thus, using immunohistofluorescence, we first evaluated protein nitrosylation levels in the ascending aorta wall. Quantitative analyses performed in the media and intima layers showed significantly higher levels of 3-NT in CD-vehicle when compared with WT littermates (*p* = 0.0008). This increase occurred in parallel with increased XOR protein levels (*p* = 0.0478) (Figure 4A,B and Appendix A). Combined CURVER treatment caused a significant decrease in aortic 3-NT and XOR levels (genotype/treatment interaction F_(1, 28)_ = 12.12, *p* = 0.0017 and F_(1, 19)_ = 5.952, *p* = 0.0247, respectively) (Figure 4A,B and Appendix A). It should be noted that the adventitia also presented this increase. Unfortunately, we did not carry out a quantitative analysis due to the difficulty of physically delimiting it, and, therefore, its detailed evaluation was not included in the present study.

Highly similar results were observed in the LV-myocardium of CD mice, with increased basal levels of 3-NT (effect of genotype, F_(1, 36)_ = 29.96, *p* < 0.0001) (Figure 4C) and XOR (effect of genotype, F_(1, 16)_ = 8.653, *p* = 0.0096) (Figure 4D), whose levels, in both cases, were normalised after CURVER treatment (effect of treatment F_(1, 36)_ = 13.10, *p* = 0.0009 and F_(1, 16)_ = 8.725, *p* = 0.0093, respectively) (Figure 4C,D). Finally, XOR protein levels were validated by western blot (Appendix A).

Detailed statistical data are shown in Appendix A.

### 2.5. CURVER Treatment Stimulates the Nuclear Translocation of Phosphorylated Form of NRF2

We previously reported that nuclear levels of NRF2 decreased in cultured primary CD cardiomyocytes with the corresponding reduced expression levels of endogenous antioxidants target genes such as Nqo1 [7]. Additionally, both CUR and VER are potent NRF2 activators [23,27]. Therefore, we next analysed the effect of CURVER treatment on nuclear levels of phosphorylated NRF2 (pNRF2) in paraffin sections of the ascending aorta and cardiac lysates of LV- myocardium.

Immunofluorescence analysis of the tunica media of ascending aortas revealed a significantly reduced number of nuclear pNRF2-positive cells in CD-vehicle mice (*p* < 0.0001) whose density were normalised after CURVER treatment (*p* = 0.5890) (Figure 5A). In agreement with previous results, we observed a significant reduction in pNRF2 protein levels in LV-myocardium lysates from CD compared with WT mice (*p* = 0.0084) (Figure 5B). CURVER treatment in CD-treated mice produced a significant increase in pNRF2 with respect to CD-vehicle mice (*p* = 0.002) (Figure 5B).

Detailed statistical analyses are shown in Appendix A.

## 3. Discussion

Cardiovascular problems are the leading cause of morbidity and mortality in patients with WBS, with a generalised arteriopathy that can be associated with systemic hypertension. Hypertension is a consistent clinical alteration among WBS patients occurring in almost 50% of children and up to 70% of adults with WBS [10,32]. Blood pressure control is thought to protect against vascular stiffness progression and consequent adverse cardiovascular events. However, no specific antihypertensive medication has been defined as the first choice in WBS. With this in mind, herein we analysed the effect of the combined treatment of curcumin (CUR), the most abundant phenol in turmeric, with verapamil (VER), a widely used voltage-dependent calcium channel blocker, on the cardiovascular phenotype of the CD murine model of WBS. The major contributions of this work could be summarised as (i) the vascular remodelling observed in WBS involves the formation and maturation of collagen fibres; (ii) xanthine oxidoreductase (XOR) is an important source of ROS that could be implicated in the cardiovascular pathology of WBS; and (iii) combined curcumin and verapamil (CURVER) therapy ameliorates the cardiovascular pathology of this disorder.

It was previously reported that both CUR and VER individually are capable of improving arterial hypertension [33,34]. In our study, both compounds partially reproduced this observation in CD mice, but their respective individual treatments caused an SBP reduction of only 10%, which is insufficient to reach the SBP values observed in WT animals. However, the combined treatment with both compounds (CURVER) fully normalised SPB values in CD mice. Concerning the heart rate, we also observed a significant effect of the treatments, although with no interaction with the genotype. A detailed analysis of the data shows no effect for CUR whilst observing a small decrease in heart rate in animals treated with VER, regardless of genotype. Unlike in WT mice, CURVER treatment in CD mice increased the heart rate reduction observed with VER alone.

The reduction in SBP was accompanied by an improvement in histopathology, both at the level of ascending aortic stenosis and cardiac hypertrophy. We observed, for the first time, a significant increase in immature collagen fibres present in the tunica media of the ascending aortas of CD animals. In accordance with previous studies implicating VER in collagen synthesis [35] and CUR in improving collagen maturation and cross-linking [36], CURVER treatment significantly reduced the increase in immature fibres observed in untreated CD mice. Therefore, CURVER treatment contributes to the remodelling of the ascending aorta, reducing the thickness of the tunica media and augmenting the diameter of the lumen. Likewise, in parallel with the improvement in aortic stenosis, we observed a reduction in cardiac hypertrophy, with a normalisation of the thickness of the LV-myocardium wall.

Another objective of our study was to better define the contribution of redox stress to the molecular physiopathology of the cardiovascular phenotype of WBS, paying special attention to XOR, which, like NADPH oxidases, is another important source of ROS in the cardiovascular system [13]. We have shown the presence of high XOR protein levels both in the ascending aorta and cardiac tissue of CD mice. These high levels correlate with increased levels of 3-NT, which are especially visible (qualitatively) in the endothelial cells (tunica intima) and also in adventitia layers. This increase in 3-NT in the aortic intima is consistent with the recent demonstration of increased nNOS/NOS1 in aortic stenosis formation in CD animals [5]. Moreover, the observed increase in 3-NT staining of the adventitia could be, in part, the consequence of an inflammatory component linked to the genetic deletion, in agreement with observations at the brain level [6]. The potential involvement of anti-inflammatory pathways deserves further investigation. CURVER treatment in CD mice normalised XOR protein levels both in the ascending aorta and LV-myocardium, as per previously published data [37]. Of interest is the observation that the reduction in XOR protein levels runs parallel to a reduction in 3-NT staining in treated CD mice.

Along this line, we previously reported that the redox stress present in cardiac tissue seems not to be fully attenuated due to the intrinsic physiological antioxidant response induced by the nuclear translocation of NRF2 [7]. We demonstrated that this response was also altered in the tunica media of the ascending aorta, with a significant reduction in nuclear pNRF2-positive cells in CD mice; an increase in the nuclear pNRF2 form, with the concomitant increase in the expression of some antioxidant response genes after treatment, could be observed in CD-treated mice. Of note, the induction of NRF2 had already been described in the case of independent CUR and VER treatments [23,27].

In conclusion, CURVER pharmacological therapy significantly mitigates the characteristic cardiovascular damage observed in a reference mouse model for WBS (CD mouse). These clinical effects are achieved via the reduction of oxidative stress levels, evidenced by the reduction of XOR protein levels and induction of the NRF2 pathway, both in the aortic wall and LV-myocardium. VER is a drug already approved for human use, while CUR is a natural, safe product. We thus propose that their combination deserves further evaluation as a potential therapeutic agent to prevent severe cardiovascular injuries occurring in human WBS patients.

## 4. Materials and Methods

### 4.1. Animals’ Maintenance

CD mice, a WBS murine model that carries a 1.3 Mb heterozygous deletion spanning from *Gtf2i* to *Fkbp6*, were obtained as previously described [4]. All mice were maintained in a 97% C57BL/6J background. Genomic DNA was extracted from a mouse ear punch to perform the genotyping using PCR and appropriate primers, as previously described [38]. Animals were housed under standard conditions in a 12 h dark/light cycle with access to food and water/treatment ad libitum.

### 4.2. Treatment Administration and Intake

Treatment was administered following the protocols previously used by the group [6]. Briefly, 10 mg/kg/day for VER treatment (equivalent to 480 mg/day in human patients) and 60 mg/kg/day CUR (Super Bio-Curcumin^®^, from Life Extension^®^, Lauderdale Lakes, FL, USA, total curcuminoids complex with essential oils of turmeric rhizome by HPLC 400 mg) were freshly prepared once per week. Both compounds were dissolved in drinking water with 1% DMSO. The vehicle control group (VEH) consisted of WT and CD mice drinking water containing 1% DMSO. Experimental groups were: 1) WT and CD mice drinking verapamil (VER), 2) WT and CD mice drinking curcumin (CUR), and 3) WT and CD mice drinking a combination of both compounds (CURVER). Treatment was started at eight weeks of age (young mice). All animals drank their treatments for four weeks before measuring arterial pressure and were then sacrificed at 12 weeks of age (young adult).

Two-way ANOVA of repeated measures did not show any significant interaction between time and treatment (Males: F_(45, 240)_ = 0.7963, *p* = 0.8189; Females: F_(45, 240)_ = 0.7335, *p* = 0.8936) (Appendix A). Also, there were no significant differences in the treatment consumption in relation to genotype (F_3, 24_ = 0.1347, *p* = 0.9384) (Appendix A).

In agreement with previous reports [4], both male and female CD mice had significantly lower body weights compared with WT mice (effect of genotype F_1, 102_ = 92.76, *p* < 0.0001 in males; F_1,62_ = 38.57, *p* < 0.0001 in females), without any significant effect of treatment (Appendix A). Appendix A shows a detailed statistical analysis.

### 4.3. Heart Rate and Arterial Pressure Measurement

The SBP and BPM of conscious mice were measured using a tail-cuff system (Non-Invasive Blood Pressure System, PanLab, Barcelona, Spain) while holding mice in a retainer tube that was cleaned after each mouse. After habituation to the retainer, measures were taken on three separate occasions and averaged per mouse.

### 4.4. Histological Preparation

Animals were perfused with 1X phosphate-buffered saline (PBS) followed by 4% paraformaldehyde (PFA). Hearts and aortas were removed and post-fixed in 4% PFA for 48 h at 4 °C and conserved in 75% EtOH until the generation of the paraffin blocks. Paraffin-embedded hearts and tissue arrays of mice aortae (TMAs) from different experimental sets were cut into 5 µm thick sections.

All measurements were carried out by two different observers blinded as to genotype and treatment. Images were captured using a Leica Leitz DMRB microscope (10x objective) equipped with a Leica DC500 camera and analysed with Fiji Image J Analysis software. For collagen analysis, samples were treated with the Picrosirius Red Staining method and analysed with a predesigned macro program [39].

### 4.5. Immunofluorescence

Paraffin-embedded cardiac and aortic tissue sections were deparaffinised and rehydrated prior to unmasking the epitope; to this end, tissue sections were treated with a retrieval solution (10 mM sodium citrate, 0.05% Tween, pH 6) for 30 min in a steamer at 95 °C. Next, sections were incubated for 20 min with ammonium chloride (NH_4_Cl, 50 mM, pH 7.4) to block free aldehyde groups, followed by a permeabilisation step (0.3% Triton x-100; 10 min), rinsed three times with PBS and then incubated with 1% BSA in PBS for 2 h at RT before overnight incubation in a humidified chamber at 4 °C with the corresponding primary antibody; anti-3-NT (1:200; Merck Millipore 06-284), anti-XOR (1:50; Rockland 200-4183S) and, anti-pNRF2 (1:200; Abcam ab76026). The following day, sections were incubated with anti-rabbit secondary antibody solutions, Alexa 555 and Alexa 647 (1:2.000, Invitrogen). Finally, sections were stained with DAPI and mounted with fluorescence mounting medium (CAT NO 0100-01; Southern Biotech) over the same glass slide.

For quantitative immunostaining analysis, four areas from each corresponding tissue section were quantified with Image J software (version 1.53). In the case of the aorta, the adventitia layer was not quantified since it was not homogeneous for all animals. All measurements were carried out in a blinded manner by two independent investigators.

### 4.6. Western Blotting

Dissected tissues were homogenised with a bullet blender and beads in RIPA buffer containing protease inhibitors (2 mM phenylmethylsulphonyl fluoride, 10 g/L aprotinin, and 1 g/L leupeptin, 1 g/L pepstatin), and phosphatase inhibitors (2 mM Na_3_VO_4_ and 100 mM NaF). Protein concentration was determined using the Dc protein assay kit (Bio-Rad). Membranes were blotted overnight at 4 °C with the following primary antibodies: anti-XDH (1:500 Santa Cruz sc-398548), anti-NRF2 (1:500 Santa Cruz, sc-H300) and anti-pNRF2 (1:5000, Abcam ab). Later, membranes were washed and incubated with the appropriate HRP-conjugated secondary antibody (1:3000 anti-rabbit W401B antibody from Promega), and the reaction was finally visualised with the Western Blotting Luminol Reagent (Santa Cruz Biotechnology). Western blot replicates were quantified using a computer-assisted densitometric analysis (Gel-Pro Analyzer, version 4; Media Cybernetics).

### 4.7. Statistical Analysis

Prior to statistical analyses, all data were analysed by the Shapiro-Wilk test to confirm normality. Male and female mice were analysed together after verifying that there were no sex differences for the parameter analysed (three-way ANOVA, sex-genotype-treatment). One- two- or three-way ANOVA with Tukey or Šídák’s post hoc test were used when needed. All data are presented as mean ± SEM. Values were considered significant when *p* < 0.05. GraphPad Prism 9 software (version 9.0.0 (121))was used for obtaining all statistical tests and graphs.

## Figures and Tables

**Figure 1 ijms-24-03261-f001:**
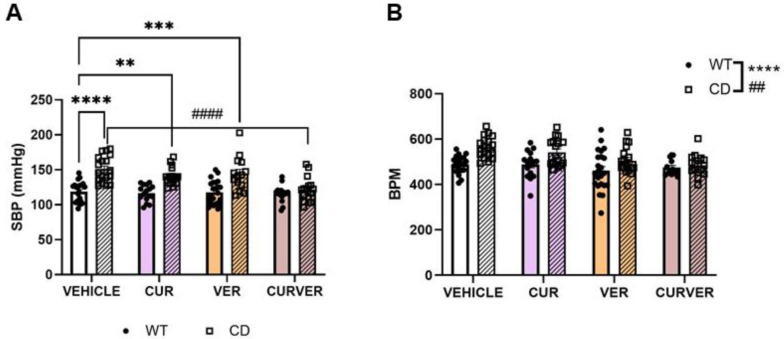
CURVER treatment normalises SBP and BPM in CD mice. (**A**) SBP and (**B**) BPM measurements in the conscious state using the indirect tail-cuff method in WT and CD mice treated with single or combined treatments. CURVER-treated CD mice showed significantly decreased SBP compared with CD-vehicle mice (*p* < 0.0001). A significant effect of genotype and treatment was observed for BPM without any significant interaction between factors (F_3, 133_ = 1.692, *p* = 0.1718). Data are represented as mean ± SEM. Two-way ANOVA with Tukey’s multiple comparisons test. * Effect of genotype; #, effect of treatment. **, ## *p* < 0.005; *** *p* < 0.001; ****, #### *p* < 0.0001.

**Figure 2 ijms-24-03261-f002:**
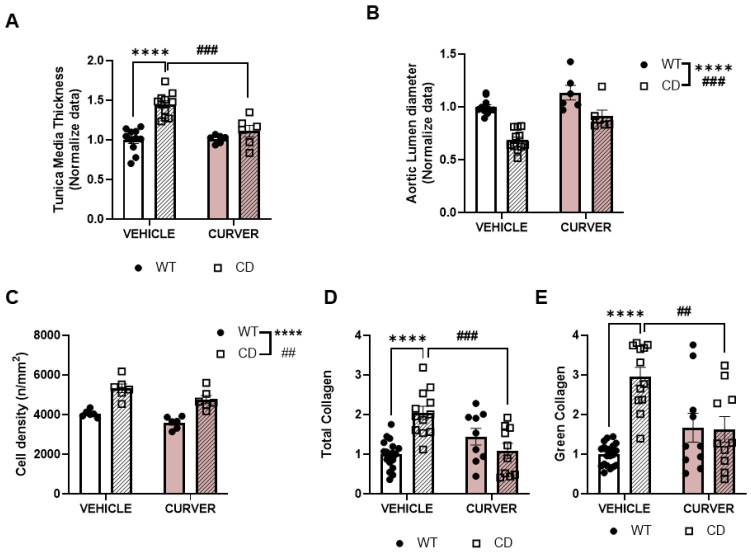
CURVER treatment prevented CD-associated ascending aortic stenosis. (**A**) Tunica media wall thickness in CURVER-treated CD mice was significantly reduced with respect to CD-vehicle (*p*= 0.0004). (**B**) A general effect of treatment could be observed in aortic lumen diameter. (**C**) A reduction in cell density, without genotype interaction, could be observed in CURVER-treated animals. (**D**) A significant genotype/treatment interaction was observed for total (F_1, 45_ = 20.81, *p* < 0.0001) and (**E**) green (F_1, 48_ = 19.20, *p* < 0.0001) collagen levels. All data were normalised to WT values. Data are represented as mean ± SEM. Two-way ANOVA with Tukey’s multiple comparisons test. * Effect of genotype; #, effect of treatment. ## *p* < 0.005; ### *p* < 0.001; **** *p* < 0.0001.

**Figure 3 ijms-24-03261-f003:**
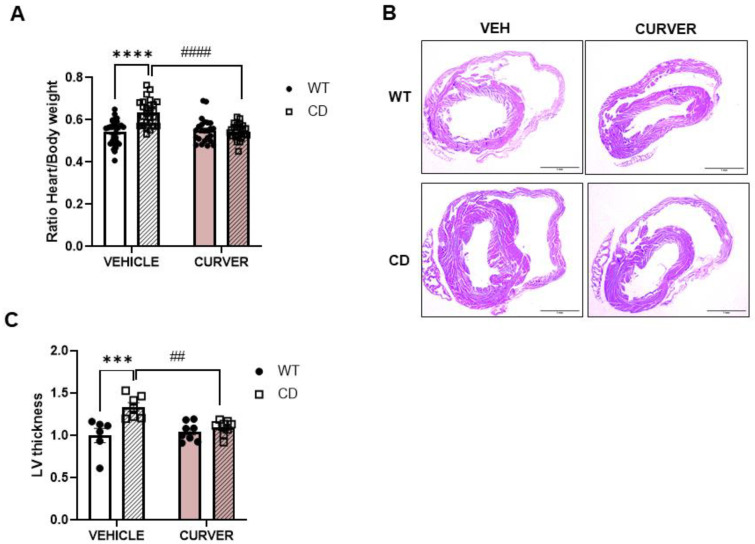
CURVER treatment attenuates cardiac hypertrophy in CD mice. (**A**) Cardiac hypertrophy (*p* < 0.0001) did not develop following CURVER administration, subsequently preventing the accompanied enhanced heart weight (*p* = 0.9995). (**B**) Representative histopathological images of hearts evaluated in (**C**) stained with haematoxylin-eosin. (**C**) The significantly increased thickening of the left ventricular wall identified in CD-vehicle mice (*p* = 0.0009) was normalised following CURVER treatment (*p* = 0.5941). Data are represented as mean ± SEM. Two-way ANOVA with Tukey’s multiple comparisons test. * Effect of genotype; #, effect of treatment. ## *p* < 0.005; *** *p* < 0.001; ****, #### *p* < 0.0001.

**Figure 4 ijms-24-03261-f004:**
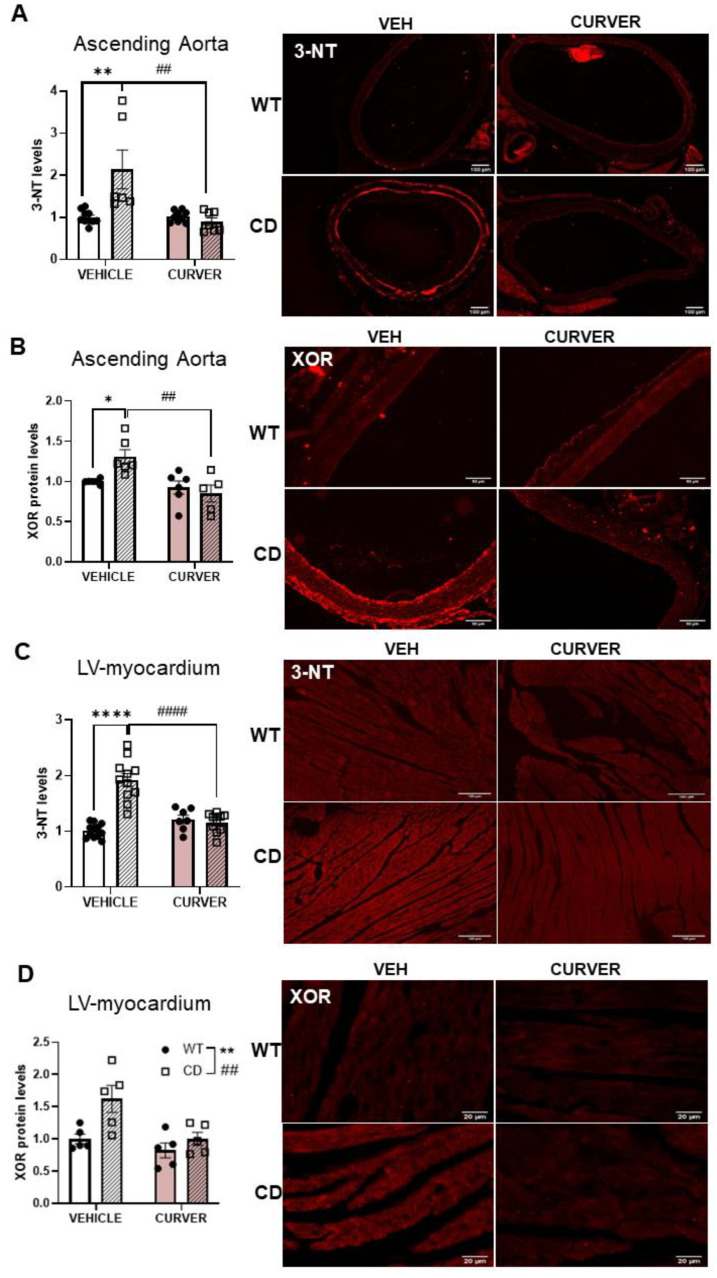
CURVER treatment reduces 3-NT and XOR aortic and cardiac levels in CD mice. (**A**) Representative images of 3-NT immunodetection in ascending aortas. Quantitative analysis showed a significant reduction in CURVER-treated versus CD-vehicle mice (*p* = 0.0007). (**B**) A significant increase in XOR protein levels occurs in the ascending aorta of CD-vehicle mice which were prevented following CURVER treatment (*p* = 0.5591). (**C**) In LV-myocardium, we observed a significant genotype/treatment interaction F_(1, 36)_ = 39.09 *p <* 0.0001) for 3-NT levels. (**D**) A significant increase in XOR protein levels of CD mice (effect of genotype, F_(1, 16)_ = 8.653, *p* = 0.0096) was reduced after CURVER treatment (effect of treatment, F_(1, 16)_ = 8.725, *p* = 0.0093). * Effect of genotype; #, effect of treatment. * *p* < 0,05; **, ## *p* < 0.005; ****, #### *p* < 0.0001.

**Figure 5 ijms-24-03261-f005:**
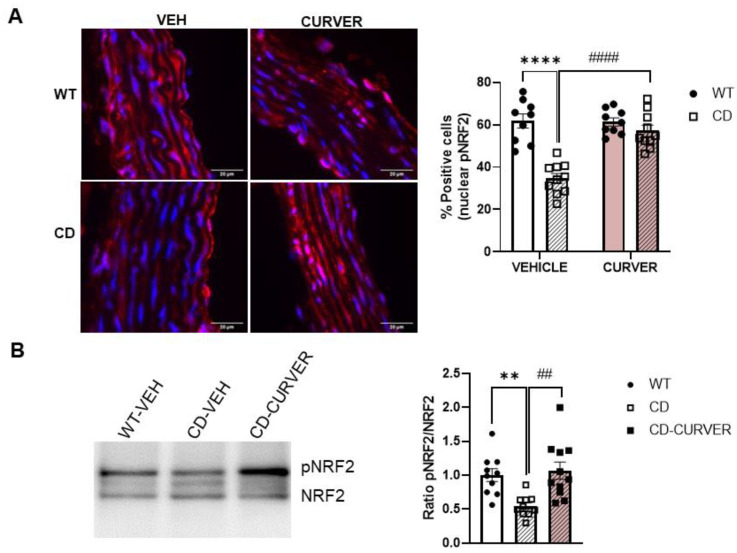
CURVER therapy increases pNRF2 nuclear location in the cardiovascular system of CD mice. (**A**) Immunohistofluorescence was performed to detect the pNRF2 protein in VSMCs of the tunica media (red) and nuclear staining (DAPI, blue) (Left panel). A significant genotype/treatment interaction could be observed (F_(1, 34)_ = 19.30; *p* = 0.0001). Two-way ANOVA with Tukey’s multiple comparisons test. (**B**) Western blot (representative image, left) showed a significant reduction in pNRF2 protein levels in lysates of CD LV-myocardium compared with WT mice (*p* = 0.0084). After CURVER treatment, CD mice showed similar amounts of pNRF2 with respect to WT mice (*p* = 0.8771). One-way ANOVA, followed by Tukey’s post hoc comparisons. Data are represented as mean ± SEM. * Effect of genotype; #, effect of treatment. **, ## *p* < 0.005; ****, #### *p* < 0.0001.

## Data Availability

Original data can be requested from the corresponding author.

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
