# Peer review of "The Combined Treatment of Curcumin with Verapamil Ameliorates the Cardiovascular Pathology in a Williams–Beuren Syndrome Mouse Model"

_ijms, 2023, doi:10.3390/ijms24043261_

Round 1

Reviewer 1 Report

The manuscript titled “The combined treatment of curcumin with verapamil ameliorates the cardiovascular 2 pathology in a Williams-Beuren syndrome mouse model” proposed the combination of curcumin and verapamil as a potential therapeutic agent for WBS. In the animal study, the authors showed that the combination resulted in normalized SPB values, which was accompanied by an improvement in histopathology at the level of ascending aortic stenosis and cardiac hypertrophy. The treatment also normalized XOR protein levels and induction of the NRF2 pathway in the ascending aorta and LV myocardium. The results indicated that the combined treatment could ameliorate the cardiovascular pathology in a WBS mouse model. Overall, this manuscript is well constructed, the data is well presented and the results are encouraging. Minor modification is needed to improve the quality of the manuscript. 

In section 4.2 Treatment administration and intake, please provide the concentration of curcumin and verapamil in water/DMSO used in the different groups. 

Author Response

First of all, thank you for the positive comments from the reviewer.
Regarding your minor changes:

Following the recommendations of both reviewers, in section 4.2 we have added the following paragraph specifying the concentrations of each compound. (lines 346 to 350)

Briefly, 10 mg/kg/day for VER treatment (equivalent to 480 mg/day in human patients) and 60 mg/kg/day CUR (Super Bio-Curcumin®, from Life Extension®, USA, total curcumoids complex with essential oils of turmeric rhizome by HPLC 400 mg ) were freshly prepared once per week. Both compounds were dissolved in drinking water with 1% DMSO.

Reviewer 2 Report

The article from N. Abdalla et al reports on the therapeutical potential of a combined treatment (CURVER) containing the phenolic compound, curcumin, and the antihypertensive drug, verapamil, against a cardiovascular disorder (Williams-Beuren syndrome) in a mice model. Particular emphasis was given to demonstrate the role of the oxidative stress in the disorder, and the improved results of the combined treatment instead of CUR and VER alone.

I think that the results are well-presented and convincing.

I therefore recommend its publication in the International Journal of Molecular Sciences after addressing the following points:

1)    It is mentioned in the introduction (lines 91-92) that „only the chronic combinatorial treatment of both compounds (CURVER), ameliorates the main cardiovascular pathology. However, the chronic versus single doses of CURVER have not been shown through the article. Have the authors collected any data concerning this point for the improvement of hypertension?

2)    The administration doses of CUR and VER are not given in Section 4.2.

Author Response

First of all, thank you for the positive comments from the reviewer.
Regarding your minor changes:

1. It is possible that, as the reviewer points out, the phrase formulated in the introduction (lines 91-92) could lead to some confusion between chronic or acute treatment. So that it does not lead to possible confusion we have eliminated "only".

Beta-blockers are first-line therapy for heart rate (HR) control in atrial fibrillation (AF) and atrial flutter (AFL)( Rajput et al.2020: J Gen Intern Med 35(12):3721–3). In this work we have not carried out any analysis of acute treatment since our aim is to propose a first choice treatment for the long-term control of SBP in patients with WBS.

2. Following the recommendations of both reviewers, in section 4.2 we have added the following paragraph specifying the concentrations of each compound (lines 346 to 350).

Briefly, 10 mg/kg/day for VER treatment (equivalent to 480 mg/day in human patients) and 60 mg/kg/day CUR (Super Bio-Curcumin®, from Life Extension®, USA, total curcumoids complex with essential oils of turmeric rhizome by HPLC 400 mg ) were freshly prepared once per week. Both compounds were dissolved in drinking water with 1% DMSO.